# ARE COLOUR TRAINED MODELS ROBUST IN HANDLING BINARY IMAGES: A FINGERPRINT RECOGNITION STUDY

## ABSTRACT

Fingerprint recognition has long been a cornerstone of biometric authentication, yet robust performance across varying imaging conditions remains a challenge, especially fingerphoto, which are generally acquired from the camera, instead of the Livescan images, which are not prone to the environmental factors. Due to the tremendous security demands in large-scale areas and areas where the deployment of computationally heavy devices might not be feasible, like refugee camps, the development of a scalable solution must be a priority. Through this research, we aim to achieve this by understanding the impact of binarization on images and models. Surprisingly, neither the role of Binarized Neural Networks (BNNs) nor binary fingerprint images (especially photos, not scans) has been explored in the literature. Henceforth, in this work, we conduct a comprehensive study of fingerprint recognition using both floating-point-based Deep Neural Networks (DNNs) and Binarised Neural Networks (BNNs) across multiple image representations, ranging from RGB to grayscale to binary. Our experiments reveal that while DNNs excel with richer representations such as RGB and grayscale, BNNs demonstrate superior compatibility with binary fingerprints, effectively leveraging their reduced complexity to achieve competitive or even better recognition accuracy. This finding highlights the importance of aligning model architectures with input spectra: full-precision networks benefit from information-rich domains, whereas binarized models coupled with binary images offer both efficiency and improved accuracy in inherently discrete representations. The results provide new insights into spectrum-aware fingerprint recognition, guiding the design of accurate and resource-efficient biometric systems.

## 1 INTRODUCTION

Fingerprint recognition has long served as the cornerstone of biometric authentication due to its uniqueness, permanence, and ease of acquisition. Over decades, research on contact-based fingerprint sensors has delivered highly reliable performance under controlled conditions, enabling deployment in large-scale national identity programs, border management, and consumer-grade authentication systems. However, with the rising demand for contactless biometric solutions—particularly in remote and mobile authentication scenarios—fingerphoto recognition, wherein fingerprints are captured using commodity cameras such as those embedded in smartphones, has emerged as a promising alternative (Donida Labati et al., 2019; Malhotra et al., 2024). This modality offers clear advantages: it is hygienic, cost-effective, and obviates the need for specialized hardware. Yet, fingerphoto recognition introduces several unique challenges arising from unconstrained acquisition: uncontrolled illumination, diverse backgrounds, variations in skin tone, motion blur, and environmental noise (Sreehari & Anzar, 2025).

Unlike contact-based sensors that are engineered to optimise ridge–valley contrast, fingerphotos are captured under consumer-grade imaging pipelines that often involve automatic white-balancing, colour processing, and compression, thereby altering the fingerprint signal. Prior works have shown that domain mismatch between contact and contactless modalities can substantially degrade recognition performance (Grosz et al., 2021). A critical but underexplored aspect of these challenges lies in the spectral representation of fingerphotos, whether they are processed in RGB, grayscale,

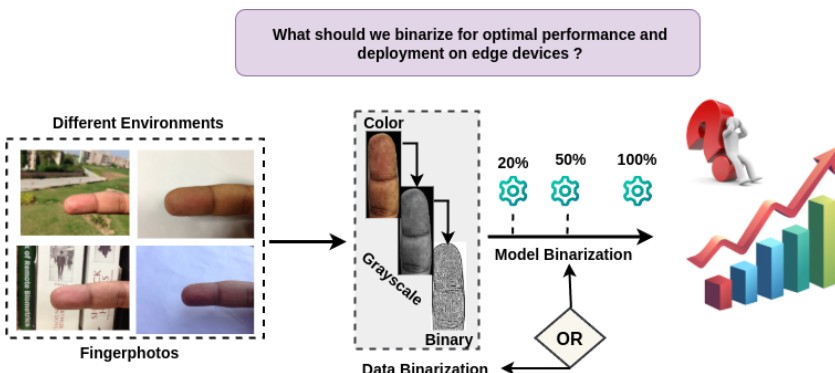

Figure 1: Can spectrum choice decide whether recognition models succeed or fail? Why do binary models thrive on binary images, while color spectra shift the outcome?

or binary form. Traditional fingerprint systems rely primarily on grayscale representations that capture ridge–valley structures and minutiae points, forming the basis of classical fingerprint matching. Fingerphotos, however, inherently capture richer colour information, which may encode additional textural cues useful for recognition (Murshed et al., 2023). Subsequently, being captured from smartphone cameras, which these days are of significantly high resolution, storing these fingerphotos is also creating a serious issue for edge devices. For example, a single fingerphoto image captured from an Android camera provides an image of resolution $3072 \times 4082$ and of size 2.51 Mb. One can imagine where millions of identities need to be matched, saving such high storage-demanding images is challenging. On the other hand, binary representations emphasise the ridge structures more explicitly, but at the cost of discarding subtle colour and intensity variations. The choice of spectrum thus directly influences recognition accuracy, model robustness, and computational efficiency. Despite its importance, a systematic study of spectrum-aware fingerphoto recognition, spanning RGB, grayscale, and binary domains, remains largely absent in the existing literature. Due to several challenges involved in the processing of fingerphoto images, especiallythe cost of string both heavy model weights (full precision floating points) and images (several of MBs), this research aims to advance fingerphoto recognition for mobile devices including edge-devices.

With the advent of Deep Neural Networks (DNNs), spectrum-rich representations (RGB and grayscale) can be effectively leveraged for extracting fine-grained intensity, texture, and structural cues. In contrast, Binarized Neural Networks (BNNs), which quantize weights and activations to binary values, cause significant efficiency gains by reducing both memory footprint and computational cost (Zhang et al., 2023). However, their performance often deteriorates when handling high-variance input distributions, such as unconstrained fingerphotos. This raises a fundamental research question: How does the spectral domain of input data impact fingerphoto recognition performance across full-precision and binarized models? Further, the recent study (Zhang et al., 2025) shows that the BNNs are highly robust in handling adversarial corruptions on large-scale datase than the full precision models. Therefore, utilising the BNNs coupled with binary images will not only save the computational cost but also ensure the developed safety-critical system is robust to adversarial perturbations. In this work, we present the first comprehensive study on spectrum-aware fingerphoto recognition across RGB, grayscale, and binary representations, systematically evaluating the trade-offs between accuracy, efficiency, and generalization (Figure: 1). Our contributions can be summarized as follows: (i) We conduct a systematic evaluation of widely used pretrained DNNs alongside pretrained BNNs, analyzing their recognition performance on fingerphotos represented in color (RGB), grayscale, and binary domains. (ii) We explore mixed-precision architectures by selectively binarizing layers within DNNs trained on color fingerphotos, and examine their impact on recognition accuracy across different spectral domains. This hybrid design highlights a new pathway for balancing accuracy and efficiency in fingerphoto recognition.

## 2 RELATED WORK

The transition from traditional contact-based fingerprint systems to touchless fingerphoto recognition has opened up promising directions for hygienic, cost-effective, and mobile-friendly authentication. However, it has also introduced unique challenges related to acquisition, spectral representen-

tation, and interoperability. Recent surveys highlight that unlike touch-based systems, fingerphotos are highly susceptible to uncontrolled illumination, background clutter, variations in pose, and scale distortion, making robust recognition a non-trivial task (Priesnitz et al., 2021; Kaplesh et al., 2025). These limitations emphasize the need for systematic investigations that go beyond conventional grayscale ridge–valley matching to fully exploit and evaluate spectral representations. To support research in this direction, several databases have been introduced. Chopra et al. released one of the first unconstrained fingerphoto datasets, explicitly capturing variations in illumination, background, and hand positioning, thereby highlighting the challenges of recognition under real-world conditions (Chopra et al., 2018). Other efforts such as ISPFDv2 and related collections attempted to capture semi-controlled fingerphoto images, yet fully unconstrained benchmarks remain scarce. In parallel, work on preprocessing and interoperability has sought to bridge the gap between contactless and legacy contact-based systems. Early contributions demonstrated that resolution normalization and rescaling of fingerphotos toward the 500 dpi standard used in contact sensors improved cross-domain matching (Kunsuk & Areekul, 2023). Building on this, CNN-based frameworks such as the multi-Siamese architecture of Lin and Kumar combined ridge and minutiae-based representations to learn domain-invariant features, significantly enhancing contactless-to-contact recognition performance (Lin & Kumar, 2018).

At the algorithmic level, deep neural networks have been central to recent advances in feature extraction. Tang et al. proposed FingerNet, a unified architecture that integrates fingerprint domain knowledge, such as orientation estimation, enhancement, and minutiae detection—within an end-to-end deep network, thereby improving robustness on noisy and latent fingerprints (Tang et al., 2017). Beyond domain-specific designs, researchers have also investigated whether ImageNet-pretrained object recognition networks can generalize to biometric tasks. Kumar and Agarwal showed that such models transfer surprisingly well to modalities including face, iris, and fingerprint, often providing competitive accuracy without requiring large task-specific datasets (Kumar & Agarwal, 2024). These findings suggest that spectrum-rich fingerphotos, particularly in RGB, could be better exploited by leveraging pretrained architectures. In parallel, efficiency has become an important consideration, as fingerphoto recognition is increasingly deployed on mobile devices. Recent developments in network binarization have shown that Binary Neural Networks (BNNs), which restrict both weights and activations to binary values, can drastically reduce model size and computational cost, making them well-suited for deployment on resource-constrained platforms (Qin et al., 2020). However, such binarization often comes with a drop in recognition accuracy, especially when applied to high-variance biometric data such as unconstrained fingerphotos. This raises open questions about how binarized and hybrid architectures perform when exposed to different spectral domains.

While prior works have significantly advanced touchless fingerprint recognition in terms of datasets, preprocessing pipelines, and architectural design, there remains a lack of systematic study on spectrum-aware fingerphoto recognition. Specifically, the implications of representing fingerphotos in color, grayscale, or binary domains, and how these interact with full-precision deep models versus binarized networks, remain largely unexplored. In this work, we address this gap by conducting a comprehensive analysis of fingerphoto recognition across RGB, grayscale, and binary spectra. We evaluate both pretrained and fine-tuned deep networks, investigate the generalization ability of spectrum-aware models, and explore mixed-precision architectures that selectively binarize layers to balance efficiency and accuracy. In doing so, our study provides new insights into the trade-offs between spectral representation, recognition performance, and computational complexity in fingerphoto recognition.

## 3 EXPERIMENTAL SETUP FOR COST-EFFECTIVE FINGERPHOTO RECOGNITION

### 3.1 FINGERPHOTO DATASETS

For our experiments, we employ the IIIT-D SmartPhone Finger-selfie Database v1 (ISPFDv1) (Sankaran et al., 2015), a fingerphoto dataset collected using smartphones. The database originally consists of 64 subjects; however, due to missing information for one subject, we utilize data from 63 subjects in our study. For each subject, eight color fingerphotos are available, corresponding to the right index and right middle fingers. These images are captured under two distinct background conditions, natural and white, across both indoor and outdoor environments, thereby ensuring diversity

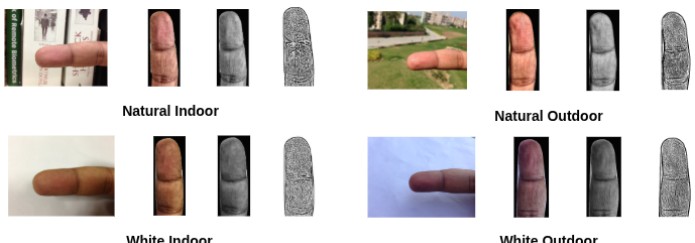

Figure 2: Figure illustrating color, grayscale, and binary fingerphoto images. It reflects the challenges involving illumination and background in fingerphoto recognition. These factors also plays major role when converting images to binary version, hence inaccraute conversion can lead to the poor performance.

in illumination and background variability. The dataset represents four acquisition scenarios: Natural Indoor (NI), Natural Outdoor (NO), White Indoor (WI), and White Outdoor (WO), each having different background conditions. It is interesting to note that livescan images are not only insensitive to illumination, but there is no role of background. Due to these factors, fingerphoto recognition is a complex case than the livescan fingerprint.

To prepare the dataset, we first segment the fingerphoto images and subsequently apply enhancement for further use. The fingerprint region is segmented using the COLFISPROOF method (Kolberg et al., 2023) with a ResNet50 backbone, followed by morphological post-processing, connected component analysis, and convex hull refinement to isolate the largest foreground region. To improve segmentation quality, we additionally generate a version of the fingerphoto with a black background, reapply the COLFISPROOF segmentation, and select the better of the two outputs. The final segmented image is then enhanced using Contrast Limited Adaptive Histogram Equalization (CLAHE) (Zuiderveld, 1994) on the luminance channel in the LAB color space. CLAHE adaptively increases local contrast while avoiding noise amplification, thereby improving ridge-valley visibility in the fingerphotos. These enhanced images are subsequently employed for fingerphoto recognition.

### 3.2 COLOR-SPACE CONVERSION: RGB-GRAY-BINARY

To normalize the spectral domain of the dataset, we convert RGB fingerphoto images into grayscale using the OpenCV implementation. Specifically, each pixel in the grayscale image is obtained as a weighted sum of the original Red, Green, and Blue channels: $I_{gray} = 0.299R + 0.587G + 0.114B$. This formulation reflects the human visual system's higher sensitivity to green, followed by red and blue, ensuring perceptually consistent brightness preservation during conversion. The process is applied uniformly across all images while maintaining the original folder hierarchy, thereby providing a consistent grayscale dataset for spectrum-aware recognition experiments.

To further normalize the spectrum, grayscale fingerphoto images are converted into binary form using *adaptive Gaussian thresholding* (Rehman & Haroon, 2023). In this approach, each pixel intensity is compared against a locally computed threshold derived from a weighted mean of its neighborhood, adjusted by a constant. Formally, for each pixel $p(x, y)$, the binary value is assigned as

$$B(x, y) = \begin{cases} 255, & \text{if } p(x, y) > T(x, y) - C, \\ 0, & \text{otherwise,} \end{cases} \quad (1)$$

where $T(x, y)$ denotes the Gaussian-weighted mean intensity of a block of size $11 \times 11$ centered at $(x, y)$, and $C = 2$ is a correction factor. This local adaptive method is more robust than global thresholding in handling illumination variations and non-uniform contrasts, thereby preserving ridge structures in the binarized fingerphoto images.

Figure 2 shows the fingerphoto images demonstrating color, grayscale, and binary spectrum, further, reflecting the information present in each spectrum.

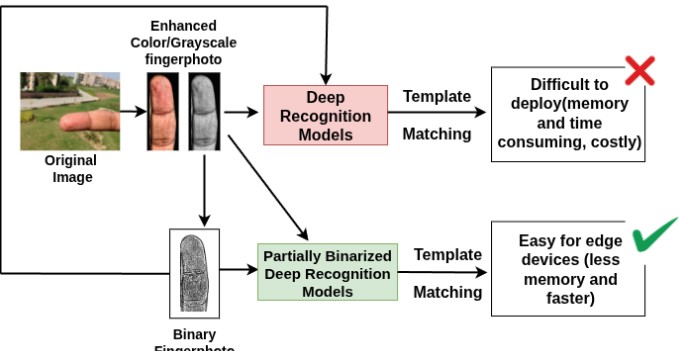

Figure 3: Figure illustrating color, grayscale, and binary fingerphoto matching. The results indicate that binary fingerphotos, when combined with partially binarized models, achieve superior recognition compared to conventional DNNs trained on color or grayscale inputs, while simultaneously reducing computational cost and model complexity.

### 3.3 ARCHITECTURES FOR FINGERPHOTO RECOGNITION

Since the biometric recognition, including fingerprint recognition, works in a zero-shot setting where testing identities are never seen during training and is inspired by the effectiveness of object recognition models (Kumar & Agarwal, 2024), the proposed research utilised several pre-trained models. In this work, we explore both full-precision and binarized deep learning models for spectrum-aware fingerphoto recognition. On the full-precision side, we consider several widely used architectures, ResNet-50, ResNet-101 (He et al., 2016), ViT-16 (Dosovitskiy et al., 2020), EfficientNet-B2, EfficientNet-B4 (Tan & Le, 2019), and ConvNeXt (Liu et al., 2022), that represent different families of convolutional and transformer-based models with proven success in large-scale image classification. To complement these, we also investigate lightweight binarized networks, specifically BNext-S, BNext-M, and BNext-L (Guo et al., 2022), which quantize weights and activations to binary values, thereby drastically reducing memory usage and computational cost (Figure 3).

Our study proceeds in two stages.

- First, we evaluate pretrained models directly on the fingerphoto dataset to establish baseline performance across RGB, grayscale, and binary domains. These models are originally trained on large-scale ImageNet variants, enabling us to assess their generalization capabilities when transferred to the biometric domain without task-specific adaptation.

- The core focus of this work lies in exploring binarization strategies. We analyze the performance of fully binarized models (BNext-S, BNext-M, BNext-L) as well as mixed-precision variants, where only selected layers of full-precision DNNs like ResNet and ViT are binarized. To fine-tune the partially binarized DNN models, we leverage the Tiny-ImageNet-200 object dataset (Wu et al., 2017). This dataset comprises approximately 1.2 million images distributed across 200 object categories. For our experiments, 500 images per class are used for training, while a separate set of 10,000 images is reserved for validation. The inclusion of Tiny-ImageNet not only provides large-scale object diversity but also enables the DNNs and binarized models to generalize effectively before being adapted for fingerphoto recognition.

  - To partially binarize ResNet-50 and ResNet-101, we replace the last 20% or 50% of convolutional layers with binarized counterparts, while keeping the earlier layers in full precision to preserve low-level representational capacity. In these binarized layers, both the convolutional weights and the input activations are quantized to binary values {-1,+1}, thereby reducing memory usage and computational overhead. Since the sign function used for binarization is non-differentiable, training is made feasible through the straight-through estimator (STE), which allows approximate gradient flow during backpropagation. To stabilize optimization and avoid loss of accuracy, each binarized convolutional layer is followed by batch normalization (BN) and dropout. This hybrid

design enables the network to retain much of the expressive power of the original full-precision ResNet while benefiting from the efficiency gains of binarization, making it well-suited for spectrum-aware fingerphoto recognition under resource constraints.

– To make BinaryViT, the last 20% or 50% of transformer layers are binarized to reduce computational complexity while retaining discriminative power. For each binary linear projection, the weight matrix is quantized as

$$\mathbf{W}_b = \text{sign}(\mathbf{W}), \quad \mathbf{W}_b \in \{-1, +1\}^{d_{\text{out}} \times d_{\text{in}}}, \tag{2}$$

while the affine-normalized input is defined as

$$\mathbf{x}' = \alpha \mathbf{x} + \beta, \tag{3}$$

, where $\alpha$, and $\beta$ are the affine transformation parameter initially taken as 1 and 0 respectively.
and the binary projection is expressed as

$$\mathbf{y} = \mathbf{x}' \mathbf{W}_b + \mathbf{b}. \tag{4}$$

In the self-attention module, binary projections are applied to the queries, keys, and values:

$$Q = \mathbf{X}\mathbf{W}_Q^b, \quad K = \mathbf{X}\mathbf{W}_K^b, \quad V = \mathbf{X}\mathbf{W}_V^b, \tag{5}$$

with the attention update computed as

$$\text{Attention}(\mathbf{X}) = \text{softmax}\left(\frac{QK^\top}{\sqrt{d_k}}\right)V. \tag{6}$$

In the feed-forward pyramid MLP, the forward pass is expressed as

$$\mathbf{H}_1 = \sigma(\mathbf{X}\mathbf{W}_1^b), \quad \mathbf{H}_p = f_{\text{pyramid}}(\mathbf{H}_1), \quad \mathbf{Y} = (\mathbf{H}_1 + \mathbf{H}_p)\mathbf{W}_2^b, \tag{7}$$

where $f_{\text{pyramid}}(\cdot)$ denotes the multi-scale convolutional aggregation. By binarizing $\mathbf{W}_Q, \mathbf{W}_K, \mathbf{W}_V, \mathbf{W}_1, \mathbf{W}_2$ while retaining full-precision normalization and residual paths, BinaryViT significantly reduces FLOPs and memory footprint, yet preserves recognition accuracy through affine scaling and residual compensation.

By systematically comparing pretrained models, fine-tuned models, and binarized models, we provide a comprehensive evaluation of how spectrum-aware representations interact with network precision in fingerphoto recognition.

## 4 FINGERPHOTO RECOGNITION RESULTS AND ANALYSIS

### 4.1 EFFECTIVENESS EVALUATION OF PRETRAINED DEEP AND BINARIZED MODELS

This study emphasizes the central question of whether color information plays a significant role in fingerphoto recognition. As shown in Table 1, when color fingerphoto images are evaluated using color-trained floating-point models such as ResNet-50 in the natural indoor (NI) scenario, the performance reaches 85.98% under both cosine and Euclidean similarity measures. Comparable results are observed with other architectures, where ResNet-101 achieves 82.80% and ViT attains 84.13% under the same conditions. Similar trends extend across the natural outdoor (NO), white indoor (WI), and white outdoor (WO) scenarios. For example, in the NO setting, ResNet-50 and ViT obtain accuracies of 85.18% and 75.40%, respectively, highlighting the consistent effectiveness of color images when used with conventional deep neural networks (DNNs). Interestingly, pretrained binary-weighted networks also demonstrate strong recognition performance on color fingerphotos. For instance, BNext-T achieves 82.80% and BNext-S yields 83.60% in NI, which are competitive with their floating-point counterparts. This trend is consistent across other acquisition scenarios, suggesting that color images provide sufficient discriminative cues regardless of whether they are processed by full-precision or binarized models. In other words, color representations appear to be robust to the underlying network precision, consistently producing compatible performance.

| Scenario | Spectrum | Metric | ResNet50 | ResNet101 | ViT | EffNet-B2 | EffNet-B4 | ConvNeXt-T | BNext-T | BNext-S | BNext-M | BNext-L |
|---|---|---|---|---|---|---|---|---|---|---|---|---|
| NI | Color | Cos | 85.98 | 82.80 | 84.13 | 80.16 | 69.58 | 73.28 | 82.80 | 83.60 | 82.80 | 78.04 |
| | | Euc | 85.98 | 82.28 | 82.28 | 80.95 | 68.52 | 73.02 | 83.33 | 83.86 | 82.54 | 77.51 |
| | Grayscale | Cos | 83.07 | 81.75 | 82.28 | 78.31 | 83.60 | 80.42 | 80.42 | 82.80 | 79.89 | 81.75 |
| | | Euc | 83.07 | 80.69 | 80.42 | 78.31 | 84.13 | 82.01 | 80.16 | 83.60 | 80.42 | 80.42 |
| | Binary | Cos | 62.96 | 59.52 | 60.32 | 61.38 | 64.29 | 69.05 | 65.61 | 70.90 | 70.37 | 67.99 |
| | | Euc | 61.90 | 58.73 | 61.11 | 61.64 | 62.96 | 68.78 | 65.08 | 70.63 | 70.11 | 68.25 |
| NO | Color | Cos | 85.18 | 85.18 | 75.40 | 78.83 | 79.63 | 83.07 | 82.80 | 82.54 | 79.89 | 77.25 |
| | | Euc | 84.13 | 84.13 | 73.01 | 78.04 | 79.10 | 83.07 | 83.07 | 82.01 | 79.1 | 76.45 |
| | Grayscale | Cos | 84.92 | 80.95 | 72.75 | 82.54 | 80.69 | 80.16 | 84.92 | 82.80 | 82.54 | 81.22 |
| | | Euc | 84.66 | 79.89 | 70.90 | 82.01 | 80.16 | 80.95 | 84.39 | 82.54 | 82.27 | 81.75 |
| | Binary | Cos | 64.55 | 66.66 | 53.97 | 69.57 | 69.05 | 71.69 | 70.37 | 73.81 | 70.37 | 69.58 |
| | | Euc | 62.96 | 66.14 | 53.17 | 70.10 | 66.93 | 70.90 | 70.63 | 74.07 | 70.63 | 69.05 |
| WI | Color | Cos | 72.22 | 73.54 | 71.16 | 69.58 | 77.51 | 72.22 | 68.52 | 67.99 | 70.10 | 70.90 |
| | | Euc | 70.90 | 71.95 | 70.37 | 70.63 | 76.72 | 71.43 | 68.78 | 67.46 | 70.37 | 71.16 |
| | Grayscale | Cos | 70.37 | 68.78 | 72.49 | 71.43 | 71.96 | 71.96 | 67.19 | 68.78 | 67.19 | 65.34 |
| | | Euc | 70.37 | 67.19 | 71.43 | 70.37 | 72.22 | 71.69 | 67.19 | 69.84 | 67.46 | 65.61 |
| | Binary | Cos | 50.79 | 52.64 | 48.94 | 48.94 | 53.97 | 52.38 | 52.64 | 52.38 | 52.64 | 44.44 |
| | | Euc | 49.20 | 52.64 | 48.41 | 48.15 | 53.97 | 52.91 | 51.85 | 53.17 | 53.70 | 43.65 |
| WO | Color | Cos | 70.63 | 71.43 | 61.64 | 67.72 | 69.58 | 73.28 | 71.43 | 73.28 | 73.54 | 60.32 |
| | | Euc | 65.87 | 70.90 | 61.11 | 66.4 | 68.62 | 73.01 | 71.69 | 73.28 | 74.07 | 60.05 |
| | Grayscale | Cos | 67.19 | 68.78 | 60.58 | 70.37 | 71.16 | 71.43 | 70.9 | 73.81 | 72.22 | 69.58 |
| | | Euc | 63.49 | 69.05 | 58.20 | 70.10 | 70.37 | 73.28 | 69.84 | 74.34 | 72.49 | 69.31 |
| | Binary | Cos | 53.17 | 55.29 | 48.15 | 61.37 | 61.11 | 60.85 | 52.64 | 57.94 | 55.82 | 54.50 |
| | | Euc | 53.17 | 53.97 | 48.15 | 59.52 | 60.32 | 60.32 | 51.85 | 57.41 | 56.08 | 55.55 |

Table 1: Performance comparison (Top-1 accuracy in %) across four scenarios (Natural Indoor, Natural Outdoor, White Indoor, White Outdoor), three spectral domains (Color, Grayscale, Binary), and two distance/similarity metrics (Cosine, Euclidean). Models are shown as columns.

| Scenario | Model | Fraction of Binarization | Color | | Grayscale | | Binary | |
|---|---|---|---|---|---|---|---|---|
| | | | Cos | Euc | Cos | Euc | Cos | Euc |
| NI | | 20% | 80.42 | 78.84 | 82.54 | 78.57 | 17.46 | 17.20 |
| | | 50% | 79.89 | 77.25 | 80.95 | 80.16 | **22.75** | **20.90** |
| NO | | 20% | 78.04 | 75.40 | 76.72 | 75.93 | 21.43 | 17.72 |
| | ResNet101 | 50% | 81.48 | 77.51 | 73.28 | 74.34 | 20.63 | **21.96** |
| WI | | 20% | 73.02 | 68.25 | 67.72 | 64.29 | 17.20 | 15.87 |
| | | 50% | 66.40 | 64.29 | 65.87 | 61.38 | **19.58** | **21.43** |
| WO | | 20% | 69.31 | 67.72 | 66.93 | 65.87 | 15.61 | 14.55 |
| | | 50% | 68.25 | 68.52 | 62.70 | 62.96 | **20.90** | **17.46** |

Table 2: Performance comparison (Top-1 accuracy in %) across four scenarios (Natural Indoor, Natural Outdoor, White Indoor, White Outdoor), three spectral domains (Color, Grayscale, Binary), and two similarity measures (Cosine, Euclidean) for partially binarized ResNet101 (20% and 50% layers). Best values in each row are highlighted in bold.

In contrast, binary fingerphoto representations reveal a different and more nuanced relationship with network architectures. While floating-point models such as ResNet-50 achieve at most 60.96% accuracy on binary images, binarized models exhibit notably higher performance. For example, in NI, BNext variants achieve between 60.96% and 70.59%, clearly surpassing the floating-point baselines. This effect is even more pronounced in the NO setting, where BNext-S attains 73.81%, representing nearly a 9% improvement over ResNet-50. Similar relative gains are also observed in the WI and WO scenarios. These results lead to two key insights. First, color fingerphotos retain discriminative information that enables consistent recognition performance across both floating-point and binarized networks. Second, binary fingerphotos align better with binarized models, where the compatibility between binary input distributions and binary network weights yields superior recognition accuracy compared to their floating-point counterparts. Together, these findings highlight that while color remains a strong and versatile representation, binary fingerphotos can be more efficiently and effectively recognized by binarized architectures, offering a unique advantage in resource-constrained biometric systems.

## 4.2 PERFORMANCE TRENDS IN PARTIALLY BINARIZED NETWORKS

Based on the results in Table 2, we propose the use of partially binarized deep neural networks (DNNs) to systematically examine the dependency of binary fingerphoto representations on network architecture. The analysis reveals distinct trends across spectral domains. For color inputs, we observe that increasing the proportion of binarized layers in a floating-point model generally leads to a decline in recognition performance. For example, when color fingerphotos are processed by ResNet-101 with 20% of its layers binarized, the performance is 80.42%, whereas this drops to 79.89% when 50% of the layers are binarized. A more pronounced decrease is observed in the white indoor (WI) scenario, where the performance falls from 73.02% (20% binarized) to 66.40% (50% binarized). These results suggest that color and grayscale fingerphotos, which carry richer spectral and textural cues, are better recognized by models that retain a larger fraction of floating-point precision, as binarization reduces the network's ability to preserve fine-grained intensity variations.

Interestingly, the opposite trend is observed for binary fingerphotos. Unlike color or grayscale images, binary representations consist primarily of discrete ridge–valley structures, which align more naturally with the quantized decision boundaries of binarized networks. For instance, in the NI scenario, recognition accuracy with ResNet-101 improves from 17.46% with 20% binarization to 22.75% with 50% binarization. Similarly, in the WI scenario under Euclidean distance, performance increases from 15.87% (20%) to 21.43% (50%). These results indicate that as the binarization level in the model increases, the network becomes more compatible with the binary input distribution, yielding improved performance. Taken together, these findings highlight a complementary relationship between image representation and network architecture. While color and grayscale fingerphotos benefit from models that retain higher floating-point precision, binary fingerphotos exhibit stronger affinity with highly binarized models. This dual behavior underscores the importance of spectrum-aware model design: the optimal degree of binarization should be carefully chosen depending on whether the input images are represented in color, grayscale, or binary form.

## 4.3 EFFICIENCY ANALYSIS: IMAGE AND MODEL SIZES

In addition to recognition, we analyse the storage efficiency of different spectral representations and model architectures. The results in Figure 4a show that binary fingerphoto images occupy substantially less storage compared to grayscale and color counterparts, making them more suitable for resource-constrained environments. A similar trend is observed for models, where partially binarized ResNet variants require less storage than their full-precision pretrained counterparts as shown in Figure 4b. These findings demonstrate the effectiveness of binarization in reducing both data and model size.

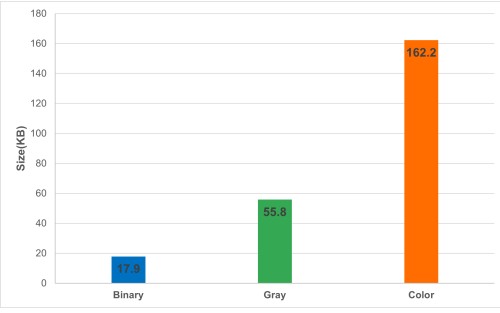
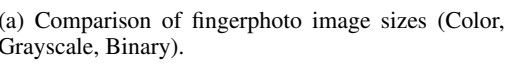

(a) Comparison of fingerphoto image sizes (Color, Grayscale, Binary).

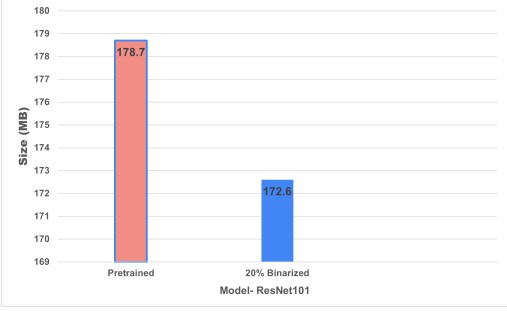

(b) Comparison of model sizes (Pretrained vs. partially binarized).

Figure 4: Storage and model size analysis. Binary fingerphoto images and partially binarized models require significantly less storage compared to their full counterparts.

### 4.4 LIMITATIONS AND FUTURE SCOPE

The dataset contains illumination variations that can affect binary fingerphoto conversion and reduce recognition accuracy. Future work can explore illumination-invariant preprocessing and more robust models.

## 5 DISCUSSION AND CONCLUSION

This study systematically examined fingerphoto recognition across different spectral representations, RGB, grayscale, and binary, using both floating-point-weights-based deep neural networks (DNNs) and binarised neural networks (BNNs). The results demonstrate a clear spectrum-dependent behaviour: full-precision DNNs achieve the best performance when processing information-rich inputs such as RGB and grayscale fingerphotos, while BNNs exhibit a stronger affinity with binary images, where the discrete ridge–valley structures align naturally with their quantized representations. Beyond recognition accuracy, we also observe significant efficiency advantages. Binary images require substantially less storage compared to grayscale and color images, and partially binarized models are consistently smaller in size than their full-precision counterparts. These findings emphasize the importance of spectrum-aware model selection and architecture design: full-precision networks are more effective in information-rich domains, whereas binarized networks not only achieve better alignment with binary inputs but also reduce storage and computational demands. Overall, this work provides valuable insights into the design of fingerprint recognition systems that balance accuracy, efficiency, and scalability, enabling robust deployment in real-world, mobile, and resource-constrained environments. We believe the presence of such a first-ever understanding can advance the deployment of fingerphoto recognition, especially for remote areas where high computational cost is still infeasible and secure the identities in refugee camps.

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
