# OpenReview forum: "Are Color Trained Models Robust in Handling Binary Images: A Fingerprint Recognition Study"
_ICLR.cc/2026/Conference — ICLR 2026 Conference Withdrawn Submission_

### Official Review · Reviewer_qHZj · 2025-10-23

**Soundness:** 2
**Presentation:** 2
**Contribution:** 2
**Rating:** 0
**Confidence:** 4

**Summary:**

The paper addresses the suitability of floating-point versus binarized DNNs for fingerprint recognition across modalities (RGB image, grayscale, binary image). They find benefit in binarizing some layers of DNNs, as it relates to storage cost and recognition performance (particularly on binary images).

While this paper addresses a practical problem of interest, I do not believe it to be a good fit for the ICLR venue, and think it would be better suited to a specialty biometrics conference or journal.

In particular, I do not see significant methodological innovation - the paper seems to merely test existing methodologies on a niche problem. Especially since there are no theoretical results, I don't think this paper would be of interest to researchers that want to design new machine learning methodologies.

That being said, I do see the merit of the study - I just don't think that ICLR is the right conference for this paper, given its specific focus on fingerprint recognition.

**Strengths:**

The paper addresses a practical problem in fingerprint recognition across different image modalities, and has a practical and easily implementable solution (DNN binarization) to improve performance.

**Weaknesses:**

There is very little methodological innovation. The problem is a very narrow problem that may not be of interest to the broad ICLR community.

**Questions:**

None

---

### Official Review · Reviewer_22cg · 2025-10-30

**Soundness:** 2
**Presentation:** 2
**Contribution:** 2
**Rating:** 2
**Confidence:** 4

**Summary:**

The study is well-motivated and the spectrum-aware angle is interesting, but the methodology/evidence has notable gaps (small single dataset; identification-only metrics; limited protocol detail), which makes some conclusions feel premature. Key claims (e.g., robustness/efficiency trade-offs and “BNNs prefer binary inputs”) are suggestive but not yet airtight given the evaluation design and scale.

**Strengths:**

1. The paper explicitly studies how input spectra (RGB/gray/binary) interact with network precision (full vs. binarized) for contactless fingerphotos—relevant to mobile/edge deployments.
2. Segmentation (COLFISPROOF with ResNet-50 + post-processing) and enhancement (CLAHE) are described concretely; grayscale and adaptive Gaussian binarization steps are specified, which supports repeatability.

**Weaknesses:**

1. All results rely on ISPFDv1 with 63 subjects, four scenarios (NI/NO/WI/WO). No cross-dataset validation, no larger or more recent fingerphoto datasets, no contact-vs-contactless interoperability experiment. This makes generalization claims fragile.
2. The paper reports Top-1 accuracy with cosine/Euclidean similarity—an identification metric—without verification metrics (ROC, EER, TAR@FAR), CMC curves, or details on gallery/probe splits and subject disjointness. Biometrics standards expect verification analyses and clearer splits; without them, it’s hard to gauge real-world utility.
3. Table 2 shows surprisingly low accuracies (≈15–23%) for partially binarized ResNet-101 on binary inputs, far below the ≈60–74% range of full-precision and BNext models in Table 1. This suggests training instability or a flawed binarization schedule; more analysis/ablations are needed.
4. The text motivates BNN robustness and safety-critical deployment but does not run any corruption/adversarial tests; claims remain speculative without empirical support.
5. Figure 4 discusses storage/model size but omits FLOPs, latency on representative hardware, or energy.

**Questions:**

1. Please detail the identification/verification protocols (gallery/probe construction, subject splits, number of trials, seeds). Add verification metrics (ROC, EER, TAR@FAR=1e-3/1e-4) and CMC curves; these are standard in fingerprint research and would solidify claims.
2. Why do partially binarized ResNet-101 models collapse on binary inputs (Table 2)? Provide ablations on: (a) which layers are binarized; (b) STE variant; (c) learning-rate/optimizer; (d) BN/Dropout placement; and (e) amount of fine-tuning on the target domain.
3. Can you evaluate on an additional fingerphoto dataset (and, ideally, cross-dataset) to validate spectrum-aware conclusions?
4. How sensitive are results to the binarization method (global Otsu vs. adaptive Gaussian vs. Sauvola) and to CLAHE parameters?

---

### Official Review · Reviewer_8x3B · 2025-10-31

**Soundness:** 2
**Presentation:** 4
**Contribution:** 1
**Rating:** 2
**Confidence:** 3

**Summary:**

This paper explores how binarized models can be used in fingerprint recognition. The reasons for this are that in low resource settings, these less compute heavy models can be useful. The paper thus explores the question of what role color plays in fingerprint recognition.

**Strengths:**

The paper is clearly written, with very clear diagrams, figures, and tables. The results (while problematic for reasons I will elaborate on) do  support the main claims.

**Weaknesses:**

This paper has several weaknesses, and I cannot recommend it gets accepted in this form. In particular:

1. The motivations are unclear to me. The authors repeatedly cite refugee camps as a location where this method would be useful. These are locations where the fingerprint recognition model would not need enormous throughput, and it is unclear to me on what hardware they are considering. Any basic smartphone can run the regular models without binarization, and even though binarization does lead to some spedup, since the regular models are not prohibitive, I do not understand the motivation. To be clear, I thinnk building more effiicent models is an important area of research, but I fail to understand this specific use case.

2. The setup of the study is unusual. The models are all being tested without any fine tuning on fingerprints, either using pretrained models on imagenet, or alternatively finetuning the binarized models on tiny imagenet. This is not how fingerprint models are usually trained: it is far more common to actually first fine tune on a fingerprint dataset.

3. More broadly, this study is not a good fit for ICLR. It is more of a model evaluation, but does not make any innovation beyond evaluating binarized models on a dataset.

**Questions:**

My main question is: why was a fingerprint dataset not used for finetuning?

---

### Official Review · Reviewer_SvYx · 2025-11-01

**Soundness:** 2
**Presentation:** 2
**Contribution:** 2
**Rating:** 0
**Confidence:** 3

**Summary:**

The paper investigates how the choice of input spectrum affects the performance of both full-precision and binarized deep neural networks (DNNs and BNNs) in the task of fingerphoto-based fingerprint recognition. Motivated by the growing need for lightweight, efficient, and scalable biometric systems suitable for mobile or resource-limited environments, the authors conduct a systematic empirical study. They use the IIIT-D Smartphone Finger-Selfie Database (ISPFDv1), consisting of 63 subjects, and process the images under four acquisition conditions (Natural Indoor, Natural Outdoor, White Indoor, and White Outdoor). The study compares several pre-trained architectures, including ResNet, ViT, EfficientNet, ConvNeXt, and binarized BNext variants. Furthermore, the authors explore partially binarized models by replacing a portion of DNN layers with binary equivalents.

**Strengths:**

1. The paper tests a wide range of model architectures and configurations under multiple environmental and spectral settings, ensuring that the comparison is thorough. The inclusion of both pretrained and hybrid (partially binarized) networks provides valuable insights into how performance scales with the degree of quantization.
2. The finding that binary images align better with binarized networks while color-rich inputs favor full-precision networks is intuitive yet empirically supported. This observation can inform practical decisions for deploying recognition systems on resource-constrained devices.

**Weaknesses:**

1. The entire contribution is based on comparing existing pretrained architectures without introducing a new algorithm, loss function, or theoretical formulation. While the empirical findings are well-presented, they do not constitute a conceptual or technical advancement. The “spectrum-aware” principle is a reasonable observation but remains descriptive rather than analytical or theoretically grounded.
2. The dataset used, ISPFDv1, contains only 63 subjects, which is too small to draw statistically significant or generalizable conclusions about model robustness.
3. The authors assert that binarized networks show “superior compatibility” and “robustness” with binary images, but the reported improvements are modest and sometimes within the range of experimental noise. The study also claims substantial efficiency benefits but reports only storage savings, without measuring runtime, latency, or energy consumption.
4. Although illumination and background variations are recognized as challenges, the paper provides no detailed analysis or mitigation strategy. There is no examination of how different thresholding methods for binarization affect accuracy, nor any exploration of adaptive preprocessing.

**Questions:**

1. How were the binarization and thresholding parameters chosen, and how sensitive are the results to these parameters?
2. Have the authors tested their “spectrum-aware” conclusions on any additional datasets, such as other touchless or multispectral fingerprint databases, to verify generalization?
3. Did the authors fine-tune BNNs specifically on binary images, or are the results based only on pretrained weights trained on natural images?
4. Can the proposed principle be extended to other biometric or image modalities (e.g., iris or palmprint), or is it domain-specific?

---

### Note · Authors · 2025-11-14

I have read and agree with the venue's withdrawal policy on behalf of myself and my co-authors.